# Ambiguity in Solving Imaging Inverse Problems with Deep-Learning-Based Operators

**DOI:** 10.3390/jimaging9070133

**Published:** 2023-06-30

**Authors:** Davide Evangelista, Elena Morotti, Elena Loli Piccolomini, James Nagy

**Affiliations:** 1Department of Mathematics, University of Bologna, 40126 Bologna, Italy; 2Department of Political and Social Sciences, University of Bologna, 40125 Bologna, Italy; elena.morotti4@unibo.it; 3Department of Computer Science and Engineering, University of Bologna, 40126 Bologna, Italy; elena.loli@unibo.it; 4Department of Mathematics, Emory University, Atlanta, GA 30322, USA; jnagy@emory.edu

**Keywords:** neural networks stability, image deblurring, deep learning, inverse problem in imaging

## Abstract

In recent years, large convolutional neural networks have been widely used as tools for image deblurring, because of their ability in restoring images very precisely. It is well known that image deblurring is mathematically modeled as an ill-posed inverse problem and its solution is difficult to approximate when noise affects the data. Really, one limitation of neural networks for deblurring is their sensitivity to noise and other perturbations, which can lead to instability and produce poor reconstructions. In addition, networks do not necessarily take into account the numerical formulation of the underlying imaging problem when trained end-to-end. In this paper, we propose some strategies to improve stability without losing too much accuracy to deblur images with deep-learning-based methods. First, we suggest a very small neural architecture, which reduces the execution time for training, satisfying a green AI need, and does not extremely amplify noise in the computed image. Second, we introduce a unified framework where a pre-processing step balances the lack of stability of the following neural-network-based step. Two different pre-processors are presented. The former implements a strong parameter-free denoiser, and the latter is a variational-model-based regularized formulation of the latent imaging problem. This framework is also formally characterized by mathematical analysis. Numerical experiments are performed to verify the accuracy and stability of the proposed approaches for image deblurring when unknown or not-quantified noise is present; the results confirm that they improve the network stability with respect to noise. In particular, the model-based framework represents the most reliable trade-off between visual precision and robustness.

## 1. Introduction

Image restoration is a discipline within the field of image processing focusing on the removal or reduction in distortions and artifacts from images. This topic is of interest in a wide range of applications, including medical imaging, satellite and aerial imaging, and digital photography. In this last case, blurring on images is quite frequent and several factors can cause it. To set some examples, Gaussian blur is caused by the diffraction of light passing through a lens and it is more prevalent in images captured with low-aperture lenses or in situations where the depth of field is shallow, whereas motion blur is due to handheld camera movements or low lighting conditions and slow shutter speeds [1,2,3]. Furthermore, noise seriously affects images; it is usually introduced by the acquisition systems.

Researchers have developed a number of algorithms for reducing blur and noise and image restoration is a very active field of research where new methods are continuously being proposed and developed. Such methodologies can be classified into two main categories: model-based and learning-based. The model-based techniques assume that the degradation process is known and it is mathematically described as an inverse problem [4]. The learning-based methods learn a map between the degraded and clean images during the training phase and use it to deblur new corrupted images [5].

### 1.1. Model-Based Mathematical Formulation

In model-based approaches, denoting by X the compact and locally connected subset of $R^{n}$ of the $x^{gt}$ ground truth sharp images, the relation between $x^{gt}\in X$ and its blurred and noisy observation $y^{\delta }$ is formulated as:(P)$$y^{\delta }=Kx^{gt}+e,$$
where *K* is the known blurring operator and e represents the noise on the image. We can say that, with very high probability, ||e||≤δ. In this setting, the goal of model-based image deblurring methods is to compute a sharp and unobstructed image x given $y^{\delta }$ and *K*, by solving the linear inverse problem. When noise is present, the problem (P) is typically reformulated into an optimization problem, where a data fit measure, namely, F, is minimized. Since the blurring operator *K* is known to be severely ill-conditioned, a regularization term R is added to the data-fidelity term F to avoid noise propagation. The resulting optimization problem is formulated as:(1)$$x^{*}=\arg\min_{x\in X}F\left(Kx,y^{\delta }\right)+\lambda R\left(x\right),$$
where λ>0 is the regularization parameter. This optimization problem can be solved using different iterative methods depending on the specific choices for F and R [1,6,7]. We remark that F is set as the least-squares function in the case of Gaussian noise, whereas the regularization function R can be tuned by the users according to the imaging properties they desire to enforce. Recently, plug-and-play techniques that plug a denoiser, usually a neural network, into an iterative procedure to solve the minimization problem have been used [8,9,10]. The value of λ can also be selected by automatic routines, image by image [11,12]. These features make model-based approaches mathematically explainable, flexible, and robust. However, a disadvantage is that the final result strongly depends on a set of parameters that are difficult to set up properly.

### 1.2. Deep-Learning-Based Formulation

In the last decade, deep learning algorithms have been emerging as good alternatives to model-based approaches. Disregarding any mathematical blurring operator, convolutional neural networks (NNs) can be trained to identify patterns characterizing blur on images, thus they can learn several kinds of blur and adapt to each specific imaging task. Large and complex convolutional neural networks, called UNet, have been proposed to achieve high levels of accuracy by automatically tuning and defining their inner filters and proper transformations for blur reduction, without needing any parameter setting [13,14,15,16]. Indeed, the possibility to process large amounts of data in parallel makes networks highly efficient for image processing tasks and prone to play a key role in the development of new and more advanced techniques in the future.

However, challenges and limitations in using neural networks are known from the literature. Firstly, it is difficult to understand and precisely interpret how they are making decisions and predictions, as they act as unexplainable black boxes mapping the input image $y^{\delta }$ towards $x^{gt}$ directly. Secondly, neural networks are prone to overfitting, which occurs when they become too specialized for the training samples and perform poorly on new, unseen images. Lastly, the high performance of neural networks is typically evaluated only in the so-called in-domain case, i.e., the test procedure is performed on images sharing exactly the same corruption with the training samples, hence the impact of unquantified perturbations (as noise can be) has been not widely studied yet. In other words, the robustness of NN-based image deblurring with respect to unknown noise is not guaranteed [17,18,19,20].

### 1.3. Contributions of the Article

Motivated by the poor stability but high accuracy of NN-based approaches in solving inverse imaging problems such as deblurring, this paper proposes strategies to improve stability, maintaining good accuracy, acting similarly to regularization functions in the model-based approach. Based on seeking a result showing a trade-off between stability and accuracy, we propose to use a very small neural network in place of the UNet, which is less accurate but is much more stable than larger networks. Since it has only a few parameters to identify, it consumes relatively little time and energy, thus meeting the green AI principles.

Moreover, we propose two new NN-based schemes, embedding a pre-processing step to face the network instability when solving deblurring problems, as in (P). The first scheme, denoted as FiNN, applies a model-free low-pass filter to the datum, before passing it as input to the NN. This is a good approach to be applied whenever an unknown noise is present because it does not need any model information or parameter tuning. The second scheme, called stabilized neural network (StNN), exploits an estimation of the noise statistics and the mathematical modeling of both noise and the image corruption process. Figure 1 shows a draft of the proposed frameworks, whose robustness is evaluated from a theoretical perspective and tested on an image data set.

### 1.4. Structure of the Article

The work is organized as follows. In Section 2, we formulate the NN-based action as an image reconstructor for the problem (P). In Section 3 we show our experimental setup and motivate our work on some experiments, thus, we state our proposals and derive their main properties in Section 4. Finally, in Section 5 we will report the results of some experiments to test the methods and empirically validate the theoretical analysis, before concluding with final remarks in Section 6.

## 2. Solving Imaging Inverse Problems with Deep-Learning-Based Operators

As stated in (P), image restoration is mathematically modeled as an inverse problem which derives from the discretization of Fredholm integral equations; these are ill-posed and the noise on the data is amplified in the numerically computed solution of $y^{\delta }=Kx^{gt}+e$. A rigorous theoretical analysis of the solution of such problems with variational techniques, which can be formulated as in Equation (Equation 1), has been performed, both in the continuous and discrete settings, and regularization techniques have been proposed to limit the noise spread in the solution [1,6].

To our best knowledge, a similar analysis for deep-learning-based algorithms is not present in the literature and it is quite mysterious how these algorithms behave in the presence of noise on the data. In this paper, we use some of the mathematical tools defined and proved in [20] and we propose here some techniques to limit noise spread. More details about the proposed mathematical framework in a more general setting can be found in [20].

In the following, if not differently stated, as a vector norm we consider the Euclidean norm. We first formalize the concept of the reconstructor associated with (P) with the following definition.

**Definition 1.** 
*Denoting by Rg(K) the range of K, we call $Y^{\delta }=\{y^{\delta }\in R^{n};\inf_{y\in Rg(K)}||y-y^{\delta }\left||\leq \delta \right\}$ the set of corrupted images according to δ≥0. Any continuous function $\psi :Y^{\delta }\to R^{n}$, mapping $y^{\delta }=Kx^{gt}+e$ (where ||e||≤δ with δ≥0) to an $x\in R^{n}$, is called a reconstructor.*


The associated *reconstructing error* is
(2)$$E_{\psi }\left(x^{gt},y^{\delta }\right):=||\psi \left(y^{\delta }\right)-x^{gt}\left|\right|.$$

**Definition 2.** 
*We quantify the accuracy of the reconstructor ψ, by defining the measure η>0 as:*

(3)
$$\eta =\underset{x^{gt}\in X}{\sup}||\psi \left(Kx^{gt}\right)-x^{gt}\left|\right|=\underset{x^{gt}\in X}{\sup}E_{\psi }\left(x^{gt},y^{0}\right).$$

*We say that ψ is $\eta^{-1}$-accurate [6].*


We now consider a neural network as a particular reconstructor.

**Definition 3.** 
*Given a neural network architecture A=(ν,S), where $\nu =\left(\nu_{0},\nu_{1},\cdots ,\nu_{L}\right)\in N^{L+1}$, $\nu_{L}=n$, is the width of each layer and $S=\left(S_{1,1},\cdots ,S_{L,L}\right),S_{j,k}\in R^{\nu_{j}\times \nu_{k}}$ is the set of matrices representing the skip connections, we define the parametric family $\Xi_{\theta }^{A}$ of neural network reconstructors with architecture A, parameterized by $\theta \in R^{s}$, as:*

(4)
$$\Xi_{\theta }^{A}=\left\{\psi_{\theta }:Y^{\delta }\to R^{n};\theta \in R^{s}\right\}$$

*where $\psi_{\theta }\left(y^{\delta }\right)=z^{L}$ is given by:*

(5)
$$\begin{array}{c} \left\{\begin{array}{c} z^{0}=y^{\delta }\\ z^{l+1}=\rho \left(W^{l}z^{l}+b^{l}+\sum_{k=1}^{l}S_{l,k}z^{k}\right)\;\forall l=0,\cdots ,L-1 \end{array}\right. \end{array}$$

*and $W^{l}\in R^{\nu_{l+1}\times \nu_{l}}$ is the weight matrix, and $b^{l}\in R^{\nu_{l+1}}$ is the bias vector.*


We now analyze the performance of NN-based reconstructors when noise is added to their input.

**Definition 4.** 
*Given δ≥0, the δ-stability constant $C_{\psi_{\theta }}^{\delta }$ of an $\eta^{-1}$-accurate reconstructor is defined as:*

(6)
$$C_{\psi_{\theta }}^{\delta }=\underset{\begin{array}{c} x^{gt}\in X\\ ||e||\leq \delta \end{array}}{\sup}\frac{E_{\psi }\left(x^{gt},y^{\delta }\right)-\eta }{{\left||e|\right|}_{2}}.$$



Since from Definition 4 we interestingly observe that the stability constant amplifies the noise in the data:(7)$$\left|\right|\psi_{\theta }\left(y^{0}+e\right)-{x||}_{2}\leq \eta +C_{\psi_{\theta }}^{\delta }{\left||e|\right|}_{2}\;\forall x\in X,\;\forall e\in R^{n}{,||e||}_{2}\leq \delta ,$$
with $y^{0}$ the noiseless datum, we can give the following definition:

**Definition 5.** 
*Given δ≥0, a neural network reconstructor $\psi_{\theta }$ is said to be δ-stable if $C_{\psi_{\theta }}^{\delta }\in \left[0,1\right)$.*


The next theorem states an important relation between the stability constant and the accuracy of a neural network as a solver of an inverse problem.

**Theorem 1.** 
*Let $\psi_{\theta }:R^{n}\to R^{n}$ be an $\eta^{-1}$-accurate reconstructor. Then, for any $x^{gt}\in X$ and for any δ>0, $\exists \;\tilde{e}\in R^{n}$ with $\left|\right|\tilde{e}\left|\right|\leq \delta$ such that*

(8)
$$C_{\psi_{\theta }}^{\delta }\geq \frac{\left|\right|K^{†}\tilde{e}\left|\right|-2\eta }{\left|\right|\tilde{e}\left|\right|}$$

*where $K^{†}$ is the Moore–Penrose pseudo-inverse of K.*


For the proof, see [20].

We emphasize that, even if neural networks used as reconstructors do not use any information on the operator *K*, the stability of $\psi_{\theta }$ is related to the pseudo-inverse of that operator.

## 3. Experimental Setting

Here, we describe our particular setting using neural networks as reconstructors for a deblurring application.

### 3.1. Network Architectures

We have considered three different neural network architectures for deblurring: the widely used UNet [21], the recently proposed NAFNet [22], and a green-AI-inspired 3L-SSNet [23].

The UNet and NAFNet architectures are complex, multi-scale networks, with similar overall structure but very different behavior. As shown in Figure 2, both UNet and NAFNet are multi-resolution networks, where the input is sequentially processed by a sequence of blocks $B_{1},\cdots ,B_{n_{i}}$, i=1,⋯,L and downsampled after that. After L−1 downsampling, the image is then sequentially upsampled again to the original shape through a sequence of blocks, symmetrically to what happened in the downsampling phase. At each resolution level i=1,⋯,L, the corresponding image in the downsampling phase is concatenated to the first block in the upsampling phase, to keep the information through the network. Moreover, a skip connection has also been added between the input and the output layer of the model to simplify the training, as described in [23]. The left-hand side of Figure 2 shows that the difference between UNet and NAFNet is in the structure of each block. In particular, the blocks in UNet are simple residual convolutional layers, defined as a concatenation of convolutions, ReLU, BatchNormalizations, and a skip connection. On the other side, each block in NAFNet is far more complex, containing a long sequence of gates, and convolutional and normalization layers. The key property of NAFNet, as described in [22], is that no activation function is used in the blocks, since they have been substituted by non-linear gates, thus obtaining improved expressivity and more training efficiency.

The 3-layer single-scale network (3L-SSNet) is a very simple model defined, as suggested by its name, by just three convolutional layers, each of them composed by a linear filter, followed by a ReLU activation function and a BatchNormalization layer. Since by construction the network works on single-scale images (the input is never downsampled to low-resolution level, as is common in image processing), the kernel size is crucial to increase the receptive field of the model. For this reason, we considered a 3L-SSNet with width [128,128,128] and kernel size [9×9,5×5,3×3].

### 3.2. Data Set

As a data set for our experiments we choose the widely used GoPro [24], which is composed of a large number of photographic images acquired from a GoPro camera. All the images have been cropped into 256×256 patches (with no overlapping), converted into grayscale, and normalized into [0,1]. We synthesize the blurring of each image according to (P) by considering a Gaussian corrupting effect, implemented with the 11×11 Gaussian kernel G defined as
(9)$$\begin{array}{c} G_{i,j}=\left\{\begin{array}{cc} e^{-\frac{1}{2}\frac{i^{2}+j^{2}}{\sigma_{G}^{2}}}\; & i,j\in {\left\{-5,\cdots ,5\right\}}^{2}\\ 0 & \mathrm{otherwise} \end{array}\right. \end{array}$$
with variance $\sigma_{G}=1.3$. The kernel is visualized in Figure 3, together with one of the GoPro images and its blurred counterpart.

### 3.3. Neural Networks Training and Testing

To train the neural network for deblurring, the set of available images is split into train and test subsets, with $N_{D}=2503$ and $N_{T}=1111$ images, respectively. Then, we consider a set $D={\left\{\left(y_{i}^{\delta },x_{i}^{gt}\right);\;x_{i}^{gt}\in S\right\}}_{i=1}^{N_{D}}$ for a given δ≥0. Since we set a mean squared error (MSE) loss function, an NN-based reconstructor is uniquely defined as the solution of:(10)$$\min_{\psi_{\theta }\in F_{\theta }^{A}}\sum_{i=1}^{N_{D}}\left|\right|\psi_{\theta }\left(y_{i}^{\delta }\right)-x_{i}^{gt}{\left|\right|}_{2}^{2}.$$ Each network has been trained by performing 50 epochs of the Adam optimizer with $\beta_{1}=0.9$, $\beta_{2}=0.9$, and a learning rate of $10^{-3}$. We focus on the next two experiments.

**Experiment A**. In this experiment, we train the neural networks on images only corrupted by blur (δ=0). With the aim of checking the networks’ accuracy, defined as in Section 2, we test on images without noise (*in-domain tests*). Then, to verify Theorem 1, we consider test images with added Gaussian noise, with σ=0.025 (*out-of-domain tests*).

**Experiment B**. A common practice for enforcing network stability is *noise injection* [25], consisting of training a network by adding noise components to the input. In particular, we have added a vector noise $e∼N(0,\sigma^{2}I)$, with σ=0.025. To test the stability of the proposed frameworks with respect to noise, we test with higher noise with respect to the training.

### 3.4. Robustness of the End-to-End NN Approach

Preliminary results obtained from experiment A are shown in Figure 4. The first row displays the reconstructions obtained from the in-domain tests, where we can appreciate the accuracy of all the three considered architectures. In the second row, we can see the results obtained from the out-of-domain tests, where the noise on the input data strongly corrupts the solution of the ill-posed inverse problem computed by UNet and NAFNet. Confirming what was stated by Theorem 1, the best result is obtained with the very light 3L-SSNET, which is the only NN able to handle the noise.

## 4. Improving Noise-Robustness in Deep-Learning-Based Reconstructors

As observed in Section 3, merely using a neural network to solve an inverse problem is an unstable routine. To enforce the robustness of $\psi_{\theta }$ reconstructors, we propose modifying the deep-learning-based approach by introducing a suitable operator, defined in the following as a *stabilizer*, into the reconstruction process.

**Definition 6.** 
*A continuous functions $\phi :R^{n}\to R^{n}$ is called a δ-stabilizer for a neural network reconstructor $\psi_{\theta }:R^{n}\to R^{n}$ if $\forall \;e\in R^{n}$ with ||e||≤δ, $\exists \;L_{\phi }^{\delta }\in \left[0,1\right)$ and $\exists \;e^{\prime }\in R^{n}$ with $\left|\right|e^{\prime }\left|\right|=L_{\phi }^{\delta }\left||e|\right|$ such that:*

(11)
$$\phi \left(Kx+e\right)=\phi \left(Kx\right)+e^{\prime }.$$

*In this case, the reconstructor ${\bar{\psi }}_{\theta }=\psi_{\theta }\circ \phi$ is said to be δ-stabilized. The smallest constant $L_{\phi }^{\delta }$ for which the definition holds is the stability constant $C_{\phi }^{\delta }$ of ϕ.*


Intuitively, applying a pre-processing ϕ with $L_{\phi }^{\delta }<1$ reduces the perturbation of the input data by converting a noise of amplitude bounded by δ to a corruption with norm bounded by $\delta L_{\phi }^{\delta }$. This intuition has been mathematically explained in [20]; Proposition 4.2, where a relationship between the stability constant of the stabilized reconstructor ${\bar{\psi }}_{\theta }$ and the stability constant of $\psi_{\theta }$ has been proved. In particular, if ${\bar{\psi }}_{\theta }=\psi_{\theta }\circ \phi$ is a δ-stabilized reconstructor, $L_{\psi_{\theta }}^{\delta }$, $L_{\phi }^{\delta }$ are the local Lipschitz constants of $\psi_{\theta }$ and ϕ, respectively, then:(12)$$\begin{array}{c} C_{{\bar{\psi }}_{\theta }}^{\delta }\leq L_{\psi_{\theta }}^{\delta }L_{\phi }^{\delta }. \end{array}$$ As a consequence, if $L_{\phi }^{\delta }<1$, then the stability constant of ${\bar{\psi }}_{\theta }$ is smaller than the Lipschitz constant of $\psi_{\theta }$, which implies that ${\bar{\psi }}_{\theta }$ is more stable to input perturbations.

We underline that the δ-stabilizers ϕ are effective if they preserve the characteristics and the details of the input image $y^{\delta }$. In this paper, we focus on the two following proposals of δ-stabilizers ϕ.

### 4.1. Stabilized Neural Network (StNN) Based on the Imaging Model

If the blurring operator *K* is known, it can be exploited to derive a δ-stabilizer function ϕ. We argue that information on *K* will contribute to improving the reconstruction accuracy. Specifically, we consider an iterative algorithm, converging to the solution of (Equation 1), represented by the scheme:(13)$$\left\{\begin{array}{c} x^{\left(0\right)}\in R^{n}\\ x^{\left(k+1\right)}=T_{k}\left(x^{\left(k\right)};y^{\delta }\right) \end{array}\right.$$
where $T_{k}$ is the action of the *k*-th iteration of the algorithm. Given a positive integer M∈N and a fixed starting iterate $x^{\left(0\right)}$, let us define the δ-stabilizer:(14)$$\phi_{M}\left(y^{\delta }\right)=\overset{M-1}{\underset{k=0}{◯}}T_{k}\left(x^{\left(k\right)};y^{\delta }\right).$$ By definition, $\phi_{M}$ maps a corrupted image $y^{\delta }$ to the solution computed by the iterative solver in *M* iterations.

In the following, we set as the objective function in (Equation 1) the Tikhonov-regularized least-squared function that has been analyzed in our previous work [20]:(15)$$\begin{array}{c} \arg\min_{x\in R^{n}}\frac{1}{2}||Kx-y^{\delta }{\left|\right|}_{2}^{2}+{\lambda ||x||}_{2}^{2}. \end{array}$$ In the paper, we showed that the Tikhonov-regularized formulation makes it possible to choose *M* such that $L_{\phi_{M}}^{\delta }<1$. We are aware that we could also explore different Tikhonov-like functions, such as ${\left||Dx|\right|}_{2}^{2}$ (with *D* difference operator), but this is beyond the scope of this paper. Hence, given δ and $F_{\theta }^{A}$, it is always possible to use $\phi_{M}$ as a pre-processing step, stabilizing $\psi_{\theta }$. We refer to ${\bar{\psi }}_{\theta }=\gamma_{\theta }\circ \phi_{M}$ as the *stabilized neural network* (StNN). In the numerical experiments, we use as the iterative method for the solution of (Equation 15) the conjugate gradient least squares (CGLS) iterative method [11]. Concerning the choice of the regularization parameter λ, we have chosen, for each experiment, the value minimizing the average relative error over the training set.

### 4.2. Filtered Neural Network (FiNN)

The intuition that a pre-processing step should reduce the noise present in the input data naturally leads to our second proposal, implemented by a Gaussian denoising filter. The Gaussian filter is a low-pass filter that reduces the impact of noise on the high frequencies [26]. Thus, the resulting pre-processed image is a low-frequency version of $y^{\delta }$ and the neural network $\psi_{\theta }\in F_{\theta }^{A}$ has to recover the high frequencies corresponding to the image details. Let $\phi_{G}$ represent the operator that applies the Gaussian filter to the input. We will refer to the reconstructor ${\bar{\psi }}_{\theta }=\psi_{\theta }\circ \phi_{G}$ as *filtered neural network* (FiNN).

Note that even if FiNN is employed to reduce the impact of the noise and consequently to stabilize the network solution, its $L_{\phi }^{\delta }$ constant is not smaller than one. In fact, for any $e\in R^{n}$ with ||e||≤δ, it holds:(16)$$\begin{array}{c} \phi_{G}\left(Kx+e\right)=\phi_{G}\left(Kx\right)+\phi_{G}\left(e\right) \end{array}$$
as a consequence of the linearity of $\phi_{G}$.

## 5. Results

In this section we present the results obtained in our deblurring experiments described in Section 3. To evaluate and compare the deblurred images, we use visual inspection on a selected test image and exploit the structural similarity index (SSIM) [27] on the test set. Details on the implementation can be found in the GitHub repository at https://github.com/devangelista2/Ambiguity-in-solving-Inverse-Problems, accessed on 31 May 2023.

### 5.1. Results of Experiment A

We show and comment on the results obtained in experiment A, described in Section 3.3. We remark that the aim of these tests is to measure the accuracy of the three considered neural reconstructors and of the stabilizers proposed in Section 4 and verify their sensitivity to noise in the input data. In a word, how these reconstructors handle the ill-posedness of the imaging inverse problem.

To this purpose, we visually compare the reconstructions of a single test image by the UNet and 3L-SSNet in Figure 5. The first row (which replicates some of the images of Figure 4) shows the results of the deep-learning-based reconstructors, where the out-of-domain images are clearly damaged by the noise. The FiNN and, particularly, the StNN stabilizer drastically reduce the noise, producing accurate results even for out-of-domain tests.

In order to analyze the accuracy and stability of our proposals, we compute the empirical accuracy ${\hat{\eta }}^{-1}$ and the empirical stability constant ${\hat{C}}_{\psi }^{\delta }$, respectively defined as:(17)$${\hat{\eta }}^{-1}={\left(\;\underset{x\in S_{T}}{\sup}||\psi \left(Kx\right)-{x||}_{2}\right)}^{-1}$$
and
(18)$${\hat{C}}_{\psi }^{\delta }=\underset{x\in S_{T}}{\sup}\frac{||\psi \left(Kx+e\right)-{x||}_{2}-\hat{\eta }}{{\left||e|\right|}_{2}}$$
where $S_{T}\subseteq X$ is the test set and e is a noise realization from $N(0,\sigma^{2}I)$, with ${\left||e|\right|}_{2}\leq \delta$ (different for any datum $x\in S_{T}$).

The computed values are reported in Table 1. Focusing on the estimated accuracies, the results confirm that NN is the most accurate method, followed by NAFNet and 3L-SSNet, as expected. As a consequence of Theorem 1, the values of the stability constant ${\hat{C}}_{\psi }^{\delta }$ are in reverse order: the most accurate is the least stable (note the very high value of ${\hat{C}}_{\psi }^{\delta }$ for NN!). By applying the stabilizers, the accuracy is slightly lower but the stability is highly improved (in most of the cases the constant is less than one), confirming the efficacy of the proposed solutions to handle noise and, at the same time, maintain good image quality. In particular, StNN is a stable reconstructor independent of the architecture.

To analyze the stability of the test set with respect to noise, we have plotted in Figure 6, for each test image, $E_{\psi }\left(x^{gt},y^{\delta }\right)-\hat{\eta }$ vs. ∥e∥, where the reconstruction error is defined in (Equation 2). With green and red dots we have plotted the experiments with stability constants less and greater than one, respectively, and with the blue dashed line the bisect. We note that the values reported in Table 1 for the empirical stability constant computed as supremum (see Equation (Equation 18)) are not outliers but they are representative of the results of the whole test set.

### 5.2. Results of Experiment B

In this experiment, we used noise injection in the neural networks’ training, as described in Section 3.3. This quite common strategy reduces the networks’ accuracy but improves their stability with respect to noise. However, we show that the reconstructions are not totally satisfactory when we test on out-of-domain images, i.e., when the input images are affected by noise of different intensities with respect to the training.

Figure 7 displays the reconstructions obtained by testing with both in-domain (on the left) and out-of-domain (on the right) images. Even if the NN reconstructions (column 4) are not so affected by noise as in experiment A (see Figure 4), however, noise artifacts are clearly visible, especially in UNet and NAFNet. Both the stabilizers proposed act efficiently and remove most of the noise. We observe that the restorations obtained with FiNN are smoother but also more blurred with respect to the ones computed by StNN.

An overview of the tests is displayed by the boxplots of the SSIM values sketched in Figure 8. The light blue, orange, and green boxes represent the results obtained with the NN, FiNN, and StNN methods, respectively. They confirm that the neural networks’ performance worsens with noisy data (see the different positions of light blue boxes from the left to the right column), whereas the proposed frameworks including FiNN and StNN are far more stable.

### 5.3. Analysis with Noise Varying on the Test Set

Finally, we have analyzed the performance of the methods when the input image $y^{\delta }$ is corrupted by noise ∥e∥ from $N(0,\sigma^{2}I)$, with σ varying.

In Figure 9, we plot, for one image in the test set, the absolute error between the reconstruction and the true image vs. the noise standard deviation σ. In the upper row, the results from experiment A (we remark that in this experiment we trained the networks on non-noisy data) are shown. The NN error (blue line) is out of range for very small values of σ for both UNet and NAFNet, whereas the 3L-SSNet is far more stable. In all the cases, the orange and green lines shows that FiNN and StNN improve the reconstruction error. In particular, StNN performs best in all these tests.

Concerning experiment B (in the lower row of the figure), it is very interesting to note that when the noise is smaller than the training one (corresponding to σ=0.025), the NN methods are the best performing for all the considered architectures. When σ≃0.05 the behavior changes and the stabilized methods are more accurate.

## 6. Conclusions

Starting from the consideration that the most popular neural networks used for image deblurring, such as the family of convolutional UNets, are very accurate but unstable with respect to noise in the test images, we have proposed two different approaches to obtain stability without losing too much accuracy. The first method is a very light neural architecture, called 3L-SSNET, and the second one is to stabilize the deep learning framework by introducing a pre-processing step. Numerical results on the GoPro dataset have demonstrated the efficiency and robustness of the proposed approaches under several settings, encompassing in-domain and out-of-domain testing scenarios. The 3L-SSNet overcomes UNet and NAFNet in every test where the noise on the test images exceeds the noise on the training set, combining the desired characteristics of execution speed (from a green AI perspective) and high stability. The FiNN proposal increases the stability of the NN-based restoration (the values of its SSIM do not change remarkably in all the experiments), but the restored images appear too smooth and a few small details are lost. The StNN proposal, exploiting a model-based formulation of the underlying imaging process, achieves the highest SSIM values in the most challenging out-of-domain cases, confirming its great theory-grounded potential. It represents, indeed, a good compromise between stability and accuracy. We finally remark that the proposed approach can be simply extended to other imaging applications modeled as an inverse problem, such as super-resolution, denoising, or tomography, where the neural networks learning the map from the input to the ground truth image cannot efficiently handle noise in the input data. It can be also applied to color images, by simply using a kernel *K* which defines the blur on the three RGB channels and by adapting the neural network to handle color images.

This work represents one step further in shedding light on the black-box essence of NN-based image processing.

## Figures and Tables

**Figure 1 jimaging-09-00133-f001:**
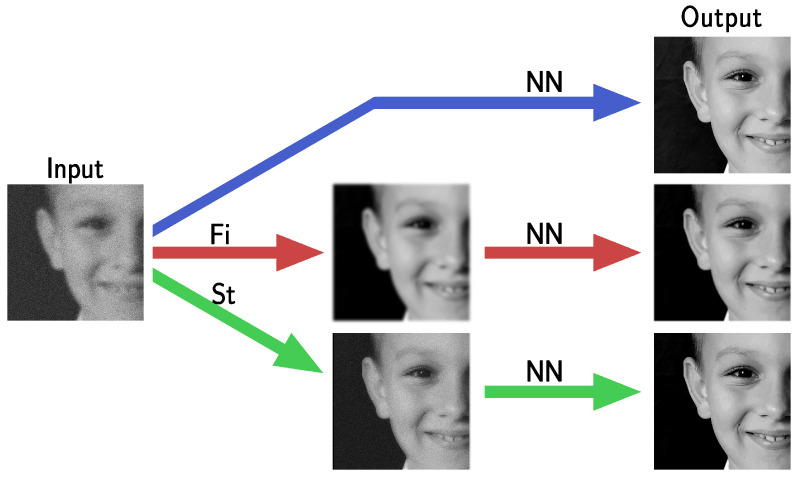
A graphical draft highlighting the introduction of pre-processing steps Fi and St defining the proposed frameworks FiNN and StNN, respectively.

**Figure 2 jimaging-09-00133-f002:**
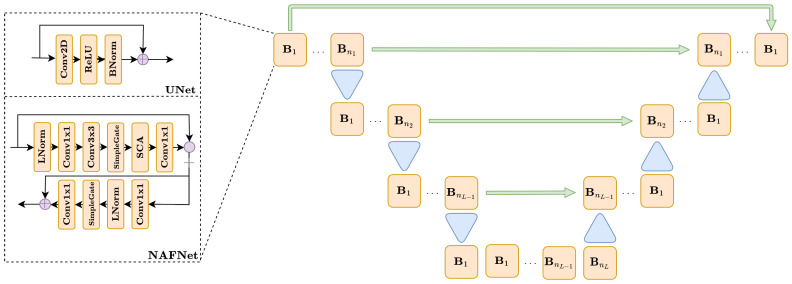
A diagram representing the UNet and NAFNet architectures.

**Figure 3 jimaging-09-00133-f003:**
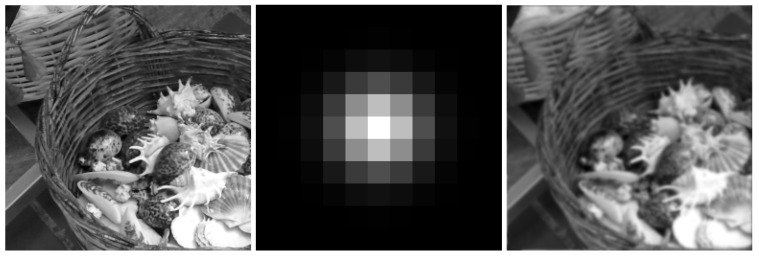
From left to right: ground truth clean image, blurring kernel, blurred corrupted image.

**Figure 4 jimaging-09-00133-f004:**
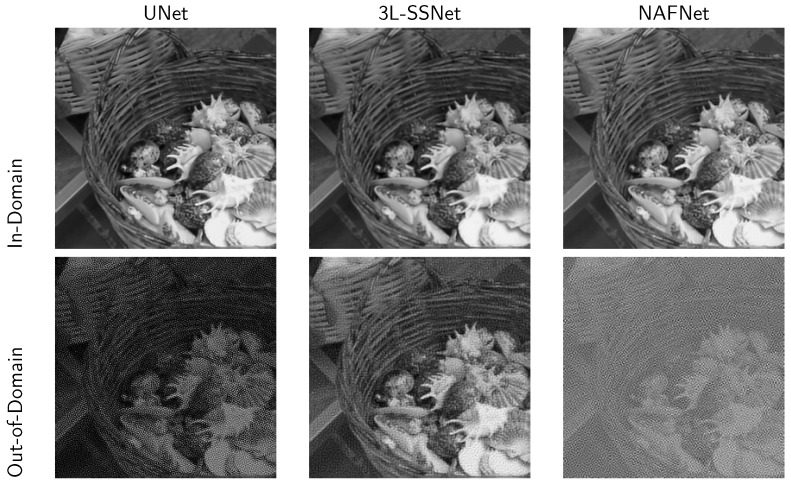
Results from experiment A with the three considered neural networks. **Upper row**: reconstruction from non-noisy data. **Lower row**: reconstruction from noisy data (δ=0.025).

**Figure 5 jimaging-09-00133-f005:**
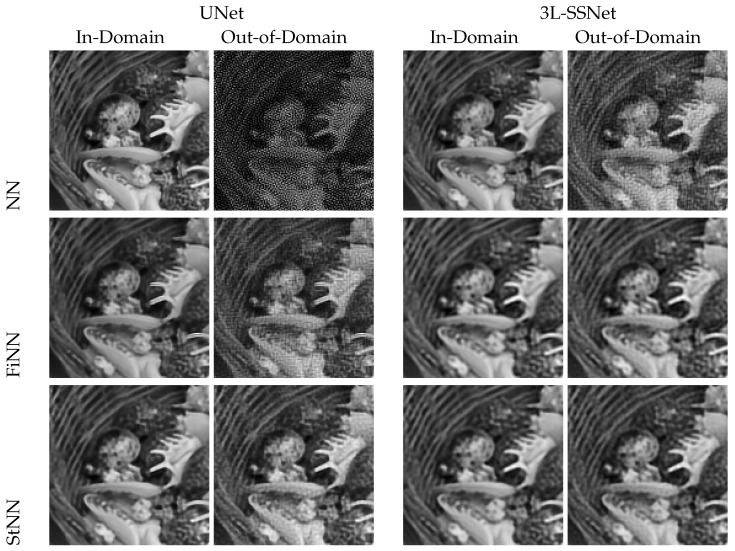
Results from experiment A with UNet and 3L-SSNet.

**Figure 6 jimaging-09-00133-f006:**
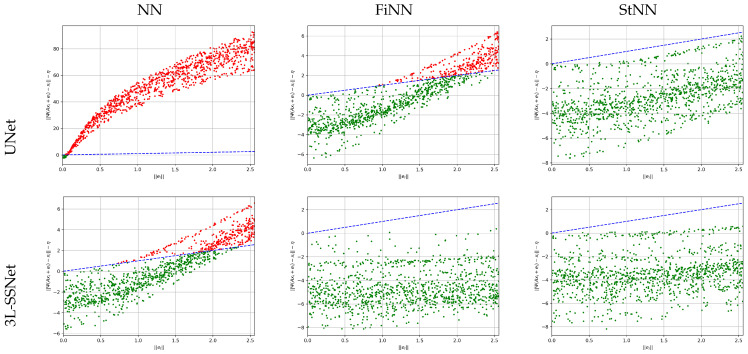
Results from experiment A. Plot of $E_{\psi }\left(x^{gt},y^{\delta }\right)-\eta$ vs. ∥e∥ for all the test images. The blue dashed line represents the bisect.

**Figure 7 jimaging-09-00133-f007:**
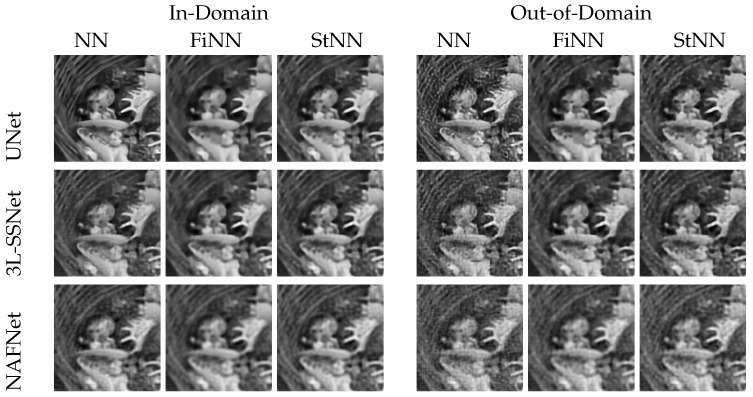
Results from experiment B. On the **left**, tests on images with the same noise as in the training (δ=0.025). On the **right**, tests on images with higher noise than in the training (δ=0.075).

**Figure 8 jimaging-09-00133-f008:**
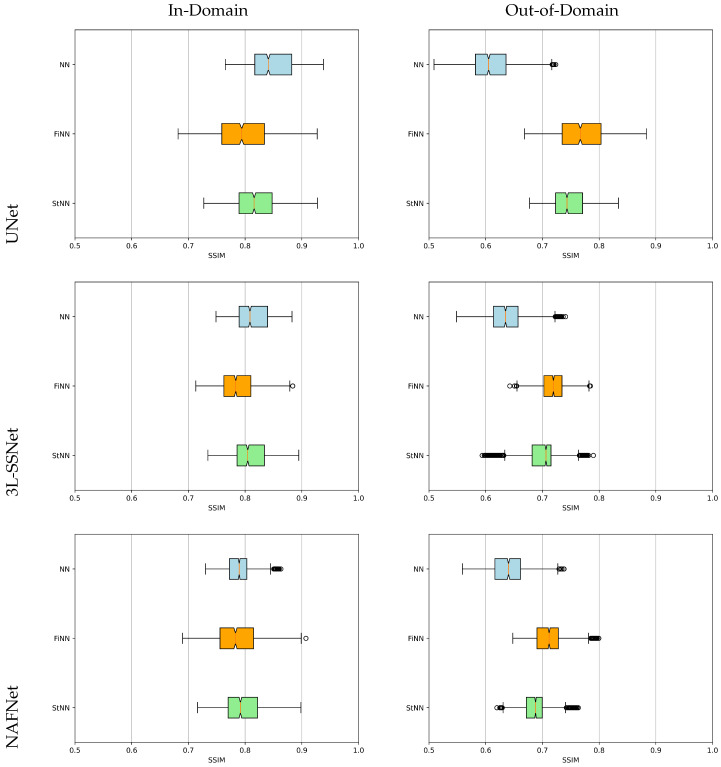
Boxplots for the SSIM values in experiment B. The light blue, orange, and green boxplots represent the results computed by NN, FiNN, and StNN, respectively.

**Figure 9 jimaging-09-00133-f009:**
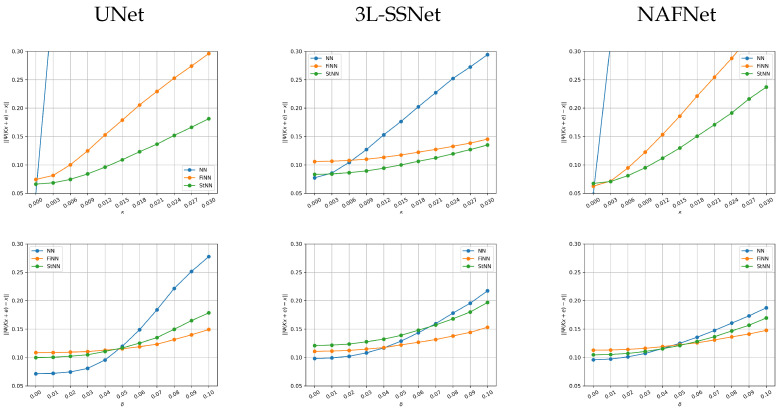
Plots of the absolute error vs. the variance σ of the noise for one image in the test set. **Upper row**: experiment A. **Lower row**: experiment B.

**Table 1 jimaging-09-00133-t001:** Estimated accuracy and stability constants for experiment A on out-of-domain test (input images corrupted by noise with δ=2.56).

	${\hat{\eta }}^{-1}$			${\hat{C}}_{\psi }^{\delta }$		
	NN	FiNN	StNN	NN	FiNN	StNN
UNet	0.118	0.085	0.087	36.572	2.519	0.878
3L-SSNet	0.082	0.055	0.072	2.563	0.148	0.243
NAFNet	0.104	0.080	0.078	15.624	1.053	0.434

## Data Availability

Not applicable.

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
