# Peer review of "Ambiguity in Solving Imaging Inverse Problems with Deep-Learning-Based Operators"

_2313-433X, 2023, doi:10.3390/jimaging9070133_

Round 1

Reviewer 1 Report

The raport on the manuscript

"Ambiguity in solving imaging inverse problems with deep learning based operators"

by Davide Evangelista, Elena Morotti, Elena Loli Piccolomini, and James Nagy

The manuscript concerns the fundamental problem of image processing which is image deblurring. Following recent trends in this area, the authors consider the approach based on deep learning. Since it is well-known that a solution to the deblurring problem is not stable with respect to noise, and this property can be also observed in a solution obtained by neural networks, the authors propose to consider some strategies to improve stability by introducing two pre-processing steps. The first one is implemented by a Gaussian denoising filter and the second one is based on the variational model with regularization. Numerical experiments are performed to verify the accuracy and stability of the proposed approaches. The results confirm that incorporation of the proposed pre-processing strategies improve the network stability with respect to noise.

The paper is well-written and organized. It refers to more thoeretical work by the same authors. The results of this work are briefly recalled and the manuscript concerns more experimental issues where the proposed pre-processing strategies are combined with different neural networks. 

The only remark that I have is the regularization term in the function (15) as it is well-known that to get the desired more regular solution, one should consider ||D x||_2^2, where D is the discrete difference operator. Maybe this should be at least commented.

Author Response

We sincerely thank the reviewer for the revision of our paper "Ambiguity in solving imaging inverse problems with deep learning based operators". 
We have corrected the paper according to your suggestions.

Question. The only remark that I have is the regularization term in the function (15) as it is well-known that to get the desired more regular solution, one should consider $||D x||_2^2$, where D is the discrete difference operator. Maybe this should be at least commented.

Answer. We agree with the reviewer that the regularization term $||D x||_2^2$ could produce more regular results. We have motivated our choice in Paragraph 4.1 now. 

Reviewer 2 Report

Review to the paper “Ambiguity in solving imaging inverse problems with deep learning based operators” by D. Evangelista, E. Morotti, E. L.
Piccolomini, and J. Nagy

In this article, the authors consider the problem of instability and poor blurring of neural network images due to their sensitivity to noise, especially when the noise on the test images exceeds the noise on the training ones. To increase the stability of this analysis, they proposed two different approaches to obtain stability without too much loss of accuracy. The first is based on the application of a very light neural architecture, and the second uses the stabilization of deep learning NN by introducing a preprocessing stage. The first overcomes the known NN results in each test, where the noise on the test images exceeds the noise on the training set, combining the desired characteristics of execution speed and high stability. At the second stage, preprocessing balances the insufficient stability of the next step based on the results of the neural network. The authors proposed two different preprocessors: the first implements a strong noise denoiser
without parameters, and the second is based on solving the corresponding inverse problem with Tikhonov regularization. Numerical experiments demonstrate the improvement of the accuracy and stability of the proposed approaches for image deblurring when unknown or not-quantified noise is present. Authors stated that this work represents one step further in shedding light on the black-box essence of NN-based image processing, and the proposed approach can be simply extended to other imaging applications modeled as an inverse problem, such as super-resolution,
denoising, for tomography, where the neural networks learning the map from the input to the ground truth image cannot efficiently handle noise in the input data.

Conclusion.
The paper is well-written; its conclusions are confirmed by results of numerical experiments. Authors presented a really new and more mathematically justified approach that combine advantages of NN and mathematical analysis that can be further developed in various mentioned ill-posed problems of physical diagnostics. I recommend to publish this paper.

Comments.
1. 1. It is necessary to describe the method for determining the regularization parameter λ in (1) and its dependence on the noise level, as well as the method for determining the  kernel of this equation K.
2. 2. Some comments on the possible application of the proposed analysis to color images are desirable.

Author Response

We sincerely thank the reviewer for the revision of our paper "Ambiguity in solving imaging inverse problems with deep learning based operators". 
We have corrected the paper according to your suggestions.

Question.  It is necessary to describe the method for determining the regularization parameter $\lambda$ in (1) and its dependence on the noise level, as well as the method for determining the  kernel of this equation K. 
Answer. Thank you for the observation. We have added a sentence about the choice of $\lambda$ in paragraph 4.1.
Regarding the kernel $K$, we have described it in paragraph 3.2. More details are in the code  available on GitHub (the link has been added in Section 5).

Question. Some comments on the possible application of the proposed analysis to color images are desirable. 
Answer. Thank you. It is very immediate to extend to color images the proposed approach, since the analysis does not depend on the number of image channels. However we have added a sentence in the conclusions about this.